# Altered Storage and Function of von Willebrand Factor in Human Cardiac Microvascular Endothelial Cells Isolated from Recipient Transplant Hearts

**DOI:** 10.3390/ijms24054553

**Published:** 2023-02-25

**Authors:** Athinoula Meli, Ann McCormack, Ianina Conte, Qu Chen, James Streetley, Marlene L. Rose, Ruben Bierings, Matthew J. Hannah, Justin E. Molloy, Peter B. Rosenthal, Tom Carter

**Affiliations:** 1Transplant Immunology, Heart Science Centre, Harefield Hospital, Hill End Road, Harefield UB9 6JH, UK; 2Molecular and Clinical Sciences Research Institute, St Georges University of London, London SW17 0RE, UK; 3Structural Biology Science Technology Platform, The Francis Crick Institute, London NW1 1AT, UK; 4Structural Biology of Cells and Viruses Laboratory, The Francis Crick Institute, London NW1 1AT, UK; 5Hematology, Erasmus University Medical Center, P.O. Box 2040, 3000 CA Rotterdam, The Netherlands; 6High Containment Microbiology, UK Health Security Agency, London NW9 5EQ, UK; 7Single Molecule Enzymology Laboratory, The Francis Crick Institute, London NW1 1AT, UK

**Keywords:** endothelial cells, von Willebrand factor, Weibel–Palade body, cardiac microvascular, secretion, exocytosis, dilated cardiomyopathy

## Abstract

The assembly of von Willebrand factor (VWF) into ordered helical tubules within endothelial Weibel–Palade bodies (WPBs) is required for the efficient deployment of the protein at sites of vascular injury. VWF trafficking and storage are sensitive to cellular and environmental stresses that are associated with heart disease and heart failure. Altered storage of VWF manifests as a change in WPB morphology from a rod shape to a rounded shape and is associated with impaired VWF deployment during secretion. In this study, we examined the morphology, ultrastructure, molecular composition and kinetics of exocytosis of WPBs in cardiac microvascular endothelial cells isolated from explanted hearts of patients with a common form of heart failure, dilated cardiomyopathy (DCM; HCMEC_D_), or from nominally healthy donors (controls; HCMEC_C_). Using fluorescence microscopy, WPBs in HCMEC_C_ (n = 3 donors) showed the typical rod-shaped morphology containing VWF, P-selectin and tPA. In contrast, WPBs in primary cultures of HCMEC_D_ (n = 6 donors) were predominantly rounded in shape and lacked tissue plasminogen activator (t-PA). Ultrastructural analysis of HCMEC_D_ revealed a disordered arrangement of VWF tubules in nascent WPBs emerging from the trans-Golgi network. HCMEC_D_ WPBs still recruited Rab27A, Rab3B, Myosin-Rab Interacting Protein (MyRIP) and Synaptotagmin-like protein 4a (Slp4-a) and underwent regulated exocytosis with kinetics similar to that seen in HCMECc. However, secreted extracellular VWF strings from HCMEC_D_ were significantly shorter than for endothelial cells with rod-shaped WPBs, although VWF platelet binding was similar. Our observations suggest that VWF trafficking, storage and haemostatic potential are perturbed in HCMEC from DCM hearts.

## 1. Introduction

Altered endothelial function is a key feature of cardiovascular and heart diseases [1]. Many of the factors that predispose to an increased risk of heart disease, such as hypertension, hypercholesterolemia, diabetes and smoking, drive cellular oxidative stress that promotes inflammation, tissue damage and tissue remodelling [2,3]. Affected blood vessels may exhibit a narrowing of the vessel lumen and impaired endothelium-dependent vasodilation, which together predispose to perturbed blood flow, tissue hypoxia and glucose deprivation. Each of these environmental stresses can trigger an adaptive protective mechanism in endothelial cells, mediated through the activation of AMP-activated protein kinase (AMPK). AMPK activation suppresses the formation of reactive oxygen species, increases nitric oxide production and reduces pro-inflammatory signalling in endothelial cells [4]. AMPK activation has recently been linked to the regulation of von Willebrand factor (VWF) trafficking and Golgi structure that directly impact the formation of Weibel–Palade bodies (WPBs), the VWF storage organelle [5]. VWF plays a crucial role in both primary and secondary hemostasis [6,7], and its physiological importance is illustrated by conditions associated with abnormal circulating levels of the protein. Low levels give rise to bleeding disorders, collectively called von Willebrand disease (VWD), but are also associated with a reduced prevalence of arterial thrombosis [8]. Elevated levels of VWF are a known risk factor for coronary heart disease, ischemic stroke and sudden death [9,10,11,12]. VWF is stored within WPBs as helical tubules that enable the packing of the protein at high concentrations [13,14,15,16,17,18]. The orderly arrangement of VWF tubules gives the WPB its distinctive rod shape [14] and is thought to be important for the correct deployment of the protein upon WPB exocytosis, whereupon it forms long (hundreds of microns), extracellular, string-like structures that efficiently capture platelets from solution under flow [19,20,21,22]. Recent studies have linked WPBs’ size and shape to the haemostatic potential of secreted VWF; short or rounded WPBs release short VWF strings, most likely due to tangling or impaired unfurling of VWF strings, resulting in a reduced haemostatic potential [23,24,25]. The discovery that AMPK activation, via environmental cues associated with cardiovascular disease, drives the formation of small WPBs has led to the suggestion that this might constitute an adaptive response to reduce the pro-thrombotic potential of the WPB under disease conditions [5,23]. 

Oxidative stress is a key feature of heart failure (HF), contributing to the pathological remodelling and dysfunction of the organ [26]. Because oxidative stress is a key trigger for the activation of the AMPK pathway, we reasoned that VWF trafficking and storage may be perturbed in microvascular endothelial cells of hearts subject to heart failure. To test this idea, we examined the morphology, ultrastructure, molecular composition and kinetics of exocytosis of WPBs in human cardiac microvascular endothelial cells (HCMEC) isolated from nominally healthy donors (HCMEC_C_) or from cardiac tissue from patients undergoing heart transplants for dilated cardiomyopathy (DCM; HCMEC_D_). To determine if the haemostatic potential of VWF was altered in HCMEC_D_, we examined the lengths of extracellular strings formed by newly secreted VWF and the ability of these strings to capture platelets out of solution under physiological flow conditions.

## 2. Results

### 2.1. WPBs in HCMEC_D_ Are Rounded in Shape

We first looked at the morphology of WPBs in HCMEC_C_ and HCMEC_D_ using immunofluorescence localization of the major WPB cargo, VWF. WPBs of HCMEC_C_ have a rod-like morphology (Figure 1(Ai,Bi)), similar to that observed in HUVEC and adult HAEC (Figure 1(Biii,iv) and Appendix A).

In contrast, HCMEC_D_ almost exclusively contained rounded organelles (Figure 1(Aii,Bii)). After staining of biopsies from recipient ventricles of explanted hearts using the WPB-specific marker, von Willebrand factor propeptide (VWFpp) [27] also showed rounded organelles within the endothelial cells of small vessels (Figure 1C, n = two donor isolates). WPBs in both HCMEC_C_ and HCMEC_D_ were positive for CD62P/P-selectin, but HCMEC_C_ also contained tPA (Appendix A). The localisation of Rab27A, Rab3B, MyRIP and Slp4-a to rounded organelles confirmed the organelles’ identity as WPBs (Figure 2) [28].

### 2.2. Ultrastructural Analysis of HCMEC_D_ WPBs Reveals Disordered VWF Tubules

To learn more about how VWF is stored in HCMEC_D_, we studied the WPB ultrastructure using transmission electron microscopy (TEM). Consistent with our immunofluorescence data (Figure 1), we found numerous rounded electron-dense membrane-bound organelles distributed within the cytoplasm (Figure 3A).

A close inspection of rounded organelles showed tangled tubule-like structures (Figure 3(Bi–iii)). Three-dimensional imaging via cryo-electron tomography (cryo-ET) of rounded granules confirmed a disordered arrangement of VWF tubules (Figure 3(Ci,ii), Appendix A), and was in stark contrast to the tubule alignment in paracrystals associated with the rod-shaped granules [14]. Ring-like structures were often observed embedded within the ball of yarn configurations (Figure 3(Di–vi)). Measurements of the inner and outer ring diameters gave values of 12.0 ± 1.0 nm and 22.3 ±.2.1 nm (s.e.m. n = 80 tubules), respectively (Figure 3(Div)), in close agreement with previous measurements of en-face VWF tubule dimensions made in situ via cryo-EM [14]. The data are consistent with the presence of rounded WPBs with VWF tubules arranged in a disordered fashion, rather than the ordered linear arrays seen in rod-shaped WPBs of HCMEC_C_ or HUVEC (Appendix A). 

### 2.3. Rounded WPBs of HCMEC_D_ form at the Trans-Golgi Network with Disordered VWF Tubules

To establish at which point tubule storage and WPB morphology become perturbed, we looked at newly forming WPBs emerging from the trans-Golgi network. In HCMEC_C_, as for HUVEC, immature WPBs emerging from the trans-Golgi network were rod-shaped, containing elongated tubules (Figure 4(Ai,ii)). In contrast, WPBs emerging from the trans-Golgi network of HCMEC_D_ were rounded in morphology, with tubules arranged in complex whorls (Figure 4(Bi–iv)) like those of the mature organelles seen in the cell periphery (Figure 3B). Evidence of organelle remodelling, a feature of normal WPB biogenesis [29], was observed (black arrow in Figure 4(Bi)). The data indicate that VWF tubule storage is perturbed during WPB biogenesis at the level of the Golgi network and not at a later stage following organelle maturation. 

### 2.4. Rounded WPBs of HCMEC_D_ Are Smaller, by Volume, Than Rod-Shaped WPBs of HCMEC_C_

Because cellular stresses associated with cardiovascular disease perturb Golgi trafficking of VWF, resulting in smaller or more rounded WPBs [5], we next asked whether rounded WPBs in HCMEC_D_ are smaller than rod-shaped WPBs in HCMEC_C._ To determine this, we estimated the volume distributions for WPBs in HCMEC_D_ and HCMEC_C_, respectively. Measurements of the long and short axes of rounded WPBs in 2D EM sections of HCMEC_D_ (e.g., Figure 5A arrows) suggest that the majority are approximately spherical in shape (Figure 5(Bi)). 

To calculate the volumes of approximately spherical WPBs, we arbitrarily selected organelles with a long-to-short axis ratio of <1.3 (337/571 organelles) (Figure 5(Bii)) and, correcting for non-diametric sectioning (see Section 4.3), estimated the organelles’ radii (Figure 5(Biii)). From this, we calculated the volume distribution for this population of organelles (Figure 5(Biv); solid line). Because it is not possible to measure organelle lengths for rod-shaped WPBs in EM thin sections, we instead measured WPB lengths from high-magnification confocal fluorescence images of VWF-immunolabeled cells and took the mean diameter of rod-shaped WPBs to be 150 nm [30]. The dotted line in Figure 5(Biv) shows the estimated volume distribution for rod-shaped WPBs in HCMEC_C_. A leftward shift in the WPB volume distribution of HCMEC_D_ (mean volume; 15.27 ± 0.26 aL (n = 1231 WPBs) compared to HCMEC_C_ (mean volume 18.16 ± 1.34 aL (n = 337 WPBs) indicates that rounded organelles are slightly smaller than rod-shaped WPBs in HCMEC_C_. 

### 2.5. WPBs in HCMEC_D_ Have a Slightly Elevated Luminal pH

The formation of rod-shaped WPBs is driven by the assembly of tubule-like structures in VWF pro-peptide dimers and the D’D3 domains of mature VWFs [13,15,17,18,31]. This process requires a low pH and high Ca environment [13,31]. To maintain tubule structure within the WPB, the organelle utilises vesicular ATPases [32,33] to maintain an acidic lumen [34]. Perturbation of intra-WPB pH results in disruption of tubule organisation and a concomitant change in organelle morphology from a rod to a rounded shape [20,35]. To establish whether rounded WPBs of HCMEC_D_ might reflect a perturbation of intra-organelle pH, we quantified the pH inside individual WPBs of VWFpp-eGFP expressing cells, as previously described [34]. The intra-WPB pH in WPBs of HCMEC_D_ was slightly, but significantly, less acidic than those of HCMEC_C_ (pH 5.7 ± 0.03 s.e.m., n = 92 WPBs, eight cells, versus 5.41 ± 0.02 s.e.m., n = 135 WPBs, seven cells, *p* < 0.0001 *t*-test, in GraphPad prism) (Figure 6). Despite the morphological perturbations, rounded WPBs recruit the key molecules required for regulated secretion (Figure 2), suggesting that the organelles would be capable of undergoing exocytosis. To examine this, we analysed the kinetics of hormone-evoked exocytosis in HCMEC_C_ and HCMEC_D_. 

### 2.6. The Kinetics of Histamine-Evoked WPB Exocytosis in HCMEC_D_ Are Similar to That in HCMEC_C_

To analyse the kinetics of histamine-evoked WPB exocytosis, we expressed VWFpp-eGFP to specifically label the WPB population in live cells [34]. Cells were also loaded with the calcium indicator Fura-2 to monitor histamine stimulation [34]. Figure 7(Ai) shows representative examples of histamine-evoked increases in [Ca^2+^]_i_, and Figure 7(Aii) shows cumulative plots of the times for individual WPB exocytosis events scaled to the fraction of the fluorescent WPB population that underwent exocytosis. In HCMEC_D_, histamine evoked a dose-dependent increase in [Ca^2+^]_i_ and WPB exocytosis. The delay in exocytosis reduced from 3.67 ± 0.39 s (s.e.m., n = 11 cells, three separate experiments) for 1 µM histamine to 2.61 ± 0.53 s at 100 µM histamine (n = 11 cells). The mean maximal rates of exocytosis were 1.42 ± 0.43 and 4.4 ± 1.4 WPB s^−1^, respectively (n = 11 cells), and the mean percentage of fluorescent WPBs that underwent exocytosis was 9.72 ± 3.14 and 19.22 ± 3.56 percent (n = 11 cells). For 100 µM histamine, the kinetics and extent of exocytosis were not significantly different from those in HCMEC_C_ (Figure 7(Aii), n = six cells from two separate isolates), indicating that the altered morphology of WPBs in HCMEC_D_ does not functionally impair regulated exocytosis.

### 2.7. VWF Strings Secreted from HCMEC_D_ Are Short Compared to Those Secreted from HUVEC although Platelet Binding per Unit Length Is No Different

Because the ordered packing of VWF tubules in rod-shaped WPBs is thought to be essential for the correct deployment of VWF into long strings on the cell surface after secretion [20], we next determined the lengths of extracellular VWF strings secreted from WPBs of HCMEC_D_. To optimise string formation, we used a parallel plate flow chamber configured to 10 Dynes/cm^2^, a value in the physiological range of venous blood flow. WPB exocytosis was triggered with histamine (100 µM) and secreted VWF strings were visualized 10–15 min after stimulation using an Alexa Fluor 594-conjugate rabbit anti-human VWF antibody added to the perfusate. Figure 8A shows the mean VWF string length observed after stimulation of HUVEC (used as a reference for cells with rod-shaped WPBs) or HCMEC_D_. In HUVEC string lengths were on average 208 ± 5.9 µm long (n = 868 strings), significantly longer than those of HCMEC_D_ (58.43 ± 1.54 µm long (n = 750 strings, *p* =< 0.0001, *t*-test, GraphPad prism)). 

Disruption to VWF tubules causing WPB rounding is reported to reduce secreted VWF string length [20]. To confirm that disruption to VWF tubules is a likely cause of the differences observed between HCMEC_D_ and HUVEC, we treated HUVEC with monensin (10 µM, 2 h). We also treated HCMEC_D_, predicting that monensin treatment would have less effect on string length due to the pre-existing VWF storage defect. Because acute treatment with monensin collapses the intra-WPB pH (Appendix A), a key driving force for VWF expulsion [34], we allowed cells to recover from monensin treatment for 16–24 h before assessing secreted VWF string length. Under these conditions, rod-shaped WPBs collapsed into rounded organelles, while the intra-WPB pH recovered close to pre-treatment levels (Appendix A). In HUVEC, VWF string length was significantly reduced after monensin treatment to values no different from those of HCMEC_D_ (Figure 8A). Monensin treatment of HCMEC_D_ resulted in a small decrease in string length (*p* = 0.019 vs. untreated, one-way ANOVA), but was not different from monensin-treated HUVEC (Figure 8A). To determine the platelet binding capacity of secreted VWF strings, we introduced platelets into the perfusate (10^8^ platelets/mL), and platelet binding was visualised using phase contrast bright field imaging. The mean number of platelets bound per unit length (1 µm) for secreted VWF strings in HCMEC_D_ was similar to that in HUVEC (Figure 8B).

## 3. Discussion

WPB formation is driven by VWF synthesis and trafficking [36]. These unusual organelles are typically rod-shaped and contain a diverse assortment of other cargo molecules, the composition of which can change in response to physiological and pathological cues [37]. Environmental cues such as oxidative stresses and fluid shear are recognised as important modulators of endothelial gene expression and function [38] and play an important role in the pathophysiology of heart failure [39]. Because such cues have recently been implicated in modulating VWF trafficking and storage in WPBs (see below [5]), we asked whether VWF storage is perturbed in HCMEC isolated from the ventricles of individuals with DCM. We found that HCMEC_D_ contains WPBs with a predominantly rounded morphology, quite distinct from the rod-shaped WPBs in cells from nominally healthy donor hearts. Not only was the morphology altered, but the cargo composition was different; tPA, a WPB cargo molecule whose expression is known to be regulated by physiological, pharmacological and environmental factors [40,41], was absent from HCMEC_D_ WPBs (Appendix A). Interestingly, patients with right-side heart failure are reported to have reduced fibrinolytic activity associated with low levels of tPA release [42]. Whether the absence of tPA in rounded WPBs reflects a co-adaption to a reduced haemostatic potential of secreted VWF remains to be established.

Our EM analysis confirmed that HCMEC_D_ WPBs are rounded in shape and contain a disordered array of VWF tubule structures (Figure 3). Several pieces of evidence indicate that the defects in WPB formation arise early in the VWF trafficking pathway and that post-Golgi maturation and trafficking of the organelle are largely unperturbed. First, nascent WPBs forming at the trans-Golgi network are already rounded and contain disrupted tubules (Figure 4). Second, rounded WPBs are smaller by volume than rod-shaped organelles (Figure 5), indicating a deficit in cargo delivery, and third, measurements of the intra-WPB pH showed that the lumen of mature organelles is highly acidic (Figure 6) and thus should be able to maintain a VWF para-crystalline assembly if formed at the trans-Golgi network. Despite an altered morphology and cargo composition (Appendix A) rounded organelles showed many features reported for typical rod-shaped WPBs: the organelles undergo membrane remodelling during biogenesis (Figure 5(Bi)), recruit the correct WPB-specific Rab proteins and effectors to regulate organelle trafficking (Figure 2), undergo luminal acidification consistent with normal maturation (Figure 6), and show evoked exocytosis with kinetics similar to WPBs in control cells (Figure 7). 

The appearance of disordered VWF tubules in WPBs still connected to the TGN suggests that the perturbation to VWF storage occurs at or before this point in the biosynthetic pathway. A key regulator of WPB size and shape is the ARF GEF, Golgi brefeldin A resistant guanine nucleotide factor 1 (GBF1), which controls the rate of VWF trafficking through the ER-Golgi interface [5]. GBF1 activity is controlled by AMPK, and AMPK, in turn, is activated by a wide range of environmental cues, many of which are closely linked to the pathology of heart failure [39]. Thus, it is attractive to speculate that AMPK activation may account in part for the perturbation of VWF trafficking seen here in HCMEC_D_. Other mechanisms may also be in operation: short or rounded WPBs have been reported in ECs exposed to perturbed fluid shear [23,43], some clinically used statins [21,22,23], and the loss of specific cellular trafficking components such as HPS6 (BLOC-2 subunit) [44] and the ER-Golgi SNAREs, SEC22B and STX5 [24,45]. Co-storage of factor VIII (FVIII) in WPBs can also cause WPB rounding [46,47,48], although we did not detect FVIII in rounded WPBs here. 

In conclusion, we have found that WPBs in HCMEC_D_, both in situ and in culture, are rounded in shape and smaller by volume than rod-shaped WPBs in control cells. Several lines of evidence are presented, indicating that these changes reflect a perturbation in VWF trafficking into the WPB. The functional consequence of altered VWF trafficking in these cells was a failure of secreted VWF to form long extracellular strings under flow conditions. So, what might be the physiological consequences of such changes? Stressors associated with heart disease and heart failure such as ischemia and hypoxia increase the risk of intravascular thrombosis. However, by modifying VWF trafficking, storage and release, these same stressors may allow endothelial cells to adapt to reduce the haemostatic potential of the secreted VWF and hence the likelihood of inappropriate coagulation [23]. Efforts are underway to explore pharmacological tools to perturb VWF trafficking and storage as a means to limit or prevent inappropriate coagulation in different disease conditions [21,22]. 

### Study Limitations

The study used cardiac endothelial cells isolated from nominally healthy individuals or DCM donors. No information about the medication status (e.g., statins) for either group was available, so we cannot exclude the possibility that statin use might account for altered VWF trafficking; however, the recruitment of Rab proteins to rod-shaped or rounded WPBs (Figure 2) suggests that rab prenylation (which is blocked by statins) was not perturbed in these cells. We had access to a limited number of samples in each group; further studies with larger sample sizes will be needed to validate and extend these findings. In situ analysis of heart tissue sections for DCM tissue showed that WPBs are rounded; however, we did not have access to tissue from the control group to determine in situ WPB morphology. Experiments were necessarily carried out in tissue culture, which does not recapitulate in vivo conditions. 

## 4. Materials and Methods

### 4.1. Tissue Culture, Transfection and Immunocytochemistry

Human umbilical vein endothelial cells (HUVEC, #C12203) were purchased from (PromoCell GmbH, Heidelberg, Germany) and cultured (maximum passage, P4) in human growth medium (HGM; Medium 199 Earle’s salts + L-Glutamine (Catalog 11150059, ThermoFisher)), supplemented with 20% fetal calf serum, 30 µg/mL endothelial cell growth supplement, 10 U/mL heparin and 50 µg/mL gentamicin at 37 °C in a 5% CO_2_ atmosphere. Adult human aortic endothelial cells (HAEC; CC-2535) and human cardiac microvascular EC from healthy individuals (control) (HCMEC_c_, CC-7030, see Appendix A for cell inventory) were purchased from Lonza Biologicals (Slough, UK) and grown in HGM. HCMEC from DCM patients (HCMEC_D_) were isolated from the ventricles of recipients’ hearts during transplantation and cultured as previously described [49,50]. Cells were nucleofected with 2–4 μg of VWF-propeptide-eGFP expression vector DNA (VWFpp-eGFP) using the Amaxa Nucleofection device™ (Lonza, Slough, UK), according to the manufacturer’s instructions [51]. Cells were plated at confluent density in complete HGM onto 35 mm diameter poly-lysine-coated glass-bottomed culture dishes (MatTeK Corp., Ashland, MA, USA) for live cell imaging of WPB exocytosis, 9 mm diameter glass coverslips for immunocytochemistry, or 35 mm coverslips for laminar flow experiments. Immunocytochemistry was performed as previously described [28]. For in situ analysis of WPB morphology, 100 µM thick ventricular tissue sections were permeabilizated using 0.3% tritonTX-100 in phosphate gelatine and saponin solution (PGAS; 0.2% (*w*/*v*) gelatin, 0.02% (*w*/*v*) saponin, 0.02% (*w*/*v*) NaN_3_, in phosphate-buffered saline) and incubated with primary antibodies for 2 h at room temperature (RT) before washing (4 times) in PGAS and incubation with secondary antibodies (2 h RT). Sections were then washed in PGAS (4 times) and mounted on standard microscope slides in Mowoil 488. Antibodies (Ab) and dilutions used are described in Appendix A of supplemental. The slides were viewed using a 63× 1.4NA oil objective on a Leica SP2 AOBS confocal microscope (Leica, Milton Keynes, UK). Images were exported to and processed in Adobe Photoshop CC release 23.2.2.

### 4.2. Transmission Electron Microscopy (TEM) and Electron Cryomicroscopy

For TEM, HCMEC were cultured for 48 h on 9 mm glass coverslips coated with 1% porcine gelatin prior to fixation with 1.5% glutaraldehyde and 2% paraformaldehyde. Post-fixation coverslips were treated with reduced OsO_4_ and tannic acid before dehydration using EtOH and propylene oxide (PO). Coverslips were then embedded “en-face” in plastic (Epon;TAAB 812), and ultrathin (55–65 nm) sections were cut parallel to cell monolayer and stained with lead citrate. Sections were viewed on a Jeol 1200EX TEM and images captured at 2000× and 20,000× magnification using an AMT XR60 camera (Deben UK, Ltd.). Images were subsequently analysed in ImageJ and Adobe Photoshop CS6/CC. For electron cryomicroscopy, HCMECs were grown on carbon film on gold grid supports for microscopy, as previously described [14]. Gold grids with cells on were washed briefly in phosphate-buffered saline (PBS), and 4 mL of 40% protein A–conjugated 10 nm gold colloid (BBI Life Sciences) in PBS was added between washing and freezing, to act as fiducial markers. Grids were frozen by plunging into liquid ethane using a manual plunge-freezer or an FEI Vitrobot Mark III (FEI Company) at room temperature and humidity (manual) or at 22 °C and room humidity (Vitrobot; humidifier switched to off). Frozen grids were stored in liquid N2. Frozen grids were imaged with an LN_2_-cooled Polara microscope (FEI Company) operating at 200 kV and equipped with an F224 HD CCD camera (TVIPS). TIA (FEI Company) and SerialEM image acquisition software were used, and low-dose procedures were used in both packages. SerialEM was used to collect whole-grid montages at 140× magnification, which were used for locating areas of interest for further imaging using low-dose procedures. Single-axis tilt-series were collected automatically using SerialEM, with an angular range of −60° to +60° and increments of 2° or 3°. Total dose for tilt series was limited to 50 to 70 e^−^/Å^2^, giving individual images with a dose of 1.2 to 1.7 e^−^/Å^2^. The dose per image was kept constant for each tilt angle in a series. The target defocus was set at −8 µm. Tomographic tilt series were aligned with the help of fiducial markers using Etomo from IMOD software [52]. Projection images in aligned tilt series were normalized based on their histograms, reconstructed to 3-dimensional volumes, and analyzed as previously described. Tomograms were denoised using IsoNet [53], then imported in Amira (v 6.4.0) (Thermofisher). Automatic tracing of VWF tubes used the ‘Cylinder Correlation’ and ‘Trace Correlation Lines’ functions in Amira, followed by manual segmentation of the WPB membrane. The Appendix A were made in Amira.

### 4.3. Estimation of the Volume of Rounded and Rod-Shaped WPBs in HCMEC_D_ and HCMEC_C_

The volume of rounded WPBs in HCMEC_D_ was estimated from measurements of organelle diameters in conventional EM sections (section thickness). To correct for the underestimate of true diameter due to non-diametric sectioning, we used an approach introduced by the co-discoverer of the WPB, Ewald Weibel, in which the true modal diameter, D_o_, is related to the apparent modal diameter, d, by $d=\pi \frac{Do}{4}$ [54,55]. Volumes for rod-shaped WPBs in HCMEC_C_ were estimated from measurements of the lengths of 1231 WPBs from confocal immunofluorescence images of 32 cells obtained from 3 separate experiments and taking a mean WPB diameter of 150 nm [30]. 

### 4.4. Live Cell Imaging of WPB Exocytosis and Intra-WPB pH

Epifluorescence imaging of changes in intracellular free calcium ion concentration ([Ca^2+^]*_i_*) and fluorescent WPB exocytosis in Fura-2 loaded (1 µM for 20 min in the dark at room temperature) endothelial cells was carried out as previously described [34]. The resting pH in VWFpp-eGFP-containing organelles in HUVEC and HCMEC was determined from epifluorescence measurements of the steady-state fluorescence of EGFP and the maximum EGFP fluorescence after acute application of the weak base ammonium chloride (NH_4_Cl; 10mM), using parameters describing the relationship between EGFP fluorescence and pH determined previously [34]. 

### 4.5. VWF String Length and Platelet Binding Density

Perfusion assays to study secreted VWF string length and platelet binding were carried out in a parallel plate flow chamber (Glycotech, Gaithersburg, MD, USA) assembled according to manufacturer’s instructions. Cells were perfused at 2 mL/minute (equivalent to 10 dyne/cm^2^) in medium M199 (ThermoFisher, Dartford, UK) supplemented with 2% bovine serum albumin (BSA). WPB exocytosis was evoked by inclusion of 100 µM histamine in the perfusate, and secreted VWF strings visualised using a polyclonal rabbit anti-human VWF antibody (1/2000, Dako North America, Carpinteria, CA, USA) pre-conjugated to Alexa Fluor 594 using the Zenon Alexa Fluor 594 rabbit IgG labelling kit (Invitrogen, Carlsbad, CA, USA) according to the manufacturer’s instructions. For analysis of platelet binding, 10^8^ platelets/mL (lyophilised platelets; Helena Biosciences, Tyne and Wear, UK) were included in the perfusate. The assembled parallel plate flow chamber was placed on the stage of Olympus IX70 inverted microscope with phase contrast optics, and time-lapse images (0.1 frame/s) were collected using the freeware imaging software WinFluor (Dr John Dempster, Strathclyde University; http://spider.science.strath.ac.uk/sipbs/showPage.php?page=software_imaging, accessed on 20 February 2023). Images were acquired for 3 min before the initiation of laminar flow and for 10–15 min during flow. A total of 10–15 consecutive optical fields were recorded along the path of flow during the recording. Endothelial cells and platelets were visualised simultaneously through phase contrast and VWF strings through fluorescence (561 nm excitation, 600–650 nm emission). For measurements of the length of VWF strings, an Olympus 20×/0.75, U Apo 340, was used, and strings shorter than 20 µm were excluded. To quantify the density of platelets per unit length of VWF strings, an Olympus 40×/1.35, UApo/340, IX70, 1-UB768 objective was used. 

### 4.6. Ethical Considerations

M.R.; The study was conducted in accordance with the Declaration of Helsinki. HCMEC_D_ were isolated and cultured from recipient hearts of DCM patients at transplant with permission of the local ethical committee, Heart Science Centre, Harefield Hospital, Hill End Road, Middlesex, UK. Ethics reference number 01-114. 

### 4.7. Statistical Analysis

Image analysis was carried out in Winfluor (http://spider.science.strath.ac.uk, accessed on 20 February 2023) or ImageJ, as previously described [32,56]. Dataset plotting, fitting and analysis were performed in Origin 2018 (OriginLab Corporation) or GraphPadPrism 9.0.2.

Data are presented as whisker box plots (Figure 3 and Figure 7) made in Origin 2018, 64-bit sr2 (Origin Labs, Northampton, MA) with parameters of: 25–75% range, inner box; mean, error bars; interquartile range (IQR) 1.5, horizontal bar; median. In-text data summaries are presented as mean ± s.e.m., n = number of observations. Statistical analysis of VWF string lengths was conducted via a one-way ANOVA and platelet binding or intra-WPB pH differences via a *t*-test in GraphPad prism [34,56].

## Figures and Tables

**Figure 1 ijms-24-04553-f001:**
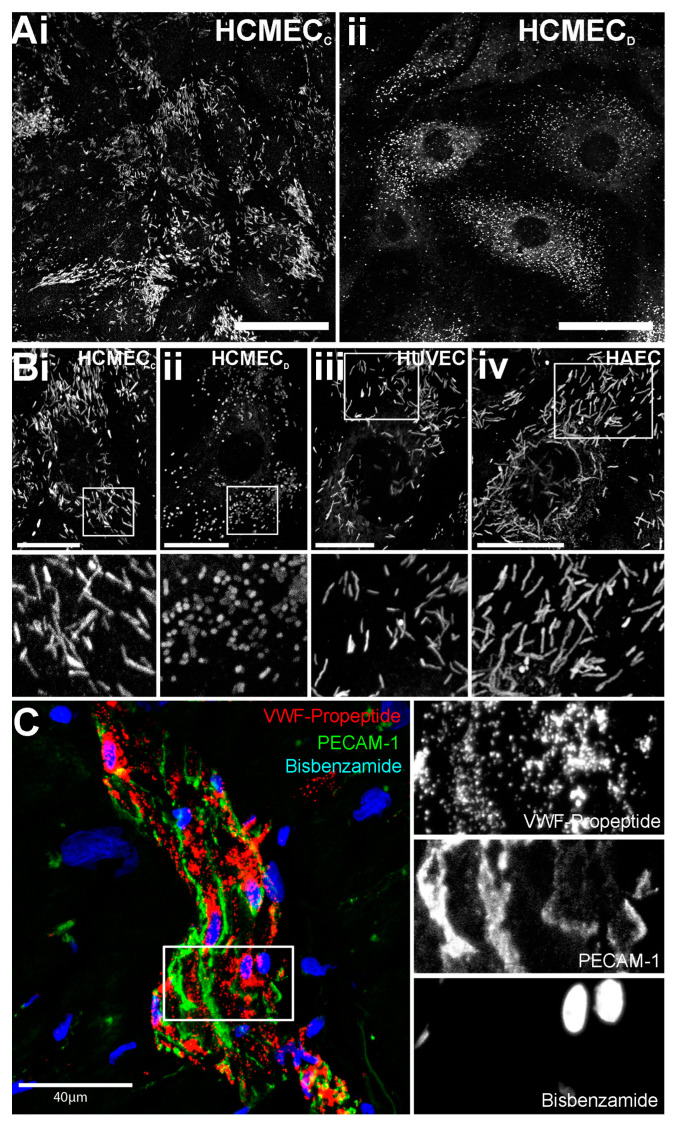
HCMEC_D_ in culture and in situ contain WPBs with a rounded morphology. Top panels in (**Ai,ii**) show representative confocal fluorescence images of (**Ai**) cultured HCMEC_C_ and (**Aii**) HCMEC_D_ immune-labelled for endogenous VWF. Top panels in (**Bi**–**iv**) show higher magnification of WPB morphology in individual (**Bi**) HCMEC_C_, (**Bii**) HCMEC_D_, HUVEC (**Biii**) and HAEC (**Biv**). Scale bars: 10 µm. The regions indicated by the white boxes in (**B**) are shown on an expanded scale in the lower panels. Images from HCMEC_C_ are representative of n = 3 separate donors, for HCMEC_D_, n = 6 donors. Images acquired using a Leica SP2 confocal microscope and Leica confocal software TCS SP2 Version 2.5 (Mannheim, Germany) equipped with 63× 1.4NA HC PL APO oil immersion objective. (**C**) shows a representative image of a 100 µm thick section of cardiac ventricle immune-labelled for endogenous VWFpp (red), PECAM-1 (green) and the nuclear stain Bisbenzamide (blue). The region indicated by the white box in (**C**) is shown on an expanded scale on the right. VWF; von Willebrand factor. HCMEC; human cardiac microvascular endothelial cells (_C_; control, _D_; dilated cardiomyopathy). HUVEC; human umbilical vein endothelial cells. HAEC; human aortic endothelial cells. VWFpp; von Willebrand factor propeptide. PECAM-1; platelet endothelial adhesion molecule-1. WPBs; Weibel–Palade bodies.

**Figure 2 ijms-24-04553-f002:**
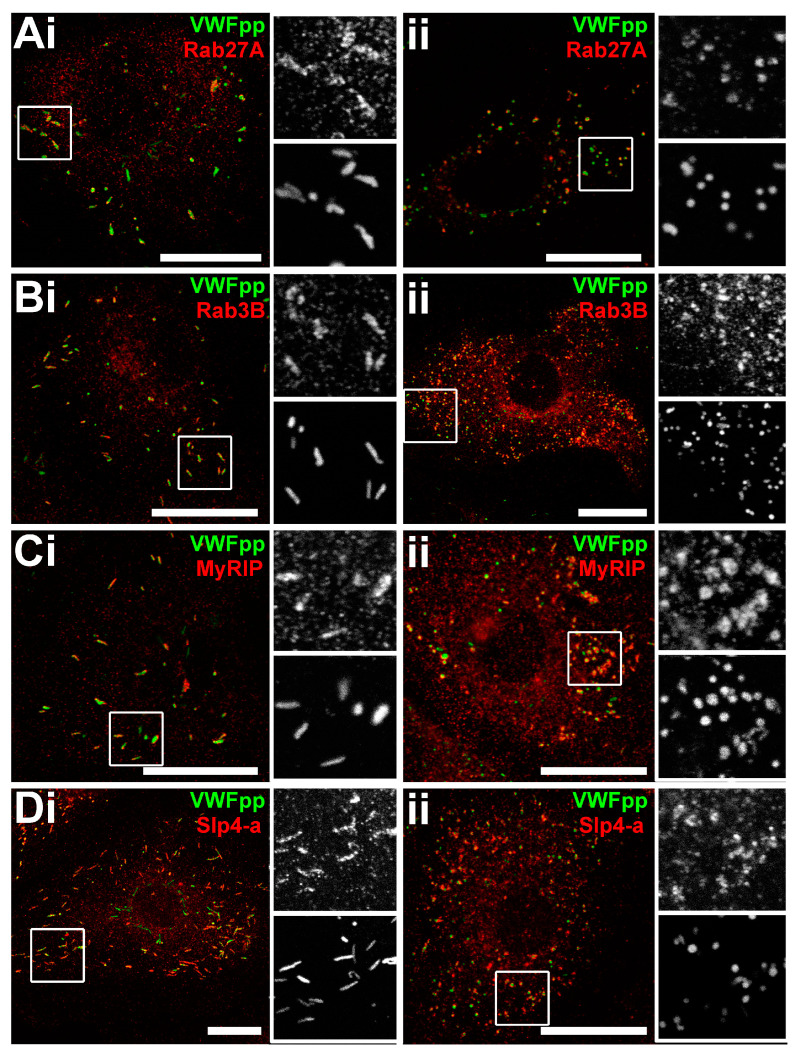
WPBs in HCMEC_C_ and HCMEC_D_ recruit endogenous Rab27A, Rab3B, MyRIP and Slp4 a. Confocal fluorescence images of HCMEC_C_ (**Ai**–**Di**) and HCMEC_D_ (**Aii**–**Dii**) immunolabeled for endogenous VWFpp (green) and in red; Rab27A (**A**), Rab3B (**B**), MyRIP (**C**) or Slp4-a (**D**), respectively. In each case, regions indicated by the white boxes are shown in grayscale on an expanded scale with VWFpp labelling in lower panels. Images are representative of 3 donors in each case. Scale bars are 20 μm. HCMEC; human cardiac microvascular endothelial cells (_C_; control, _D_; dilated cardiomyopathy). VWFpp; von Willebrand factor propeptide. MYRIP; myosin rab interacting protein. Slp4a; synaptotagmin-like protein 4a.

**Figure 3 ijms-24-04553-f003:**
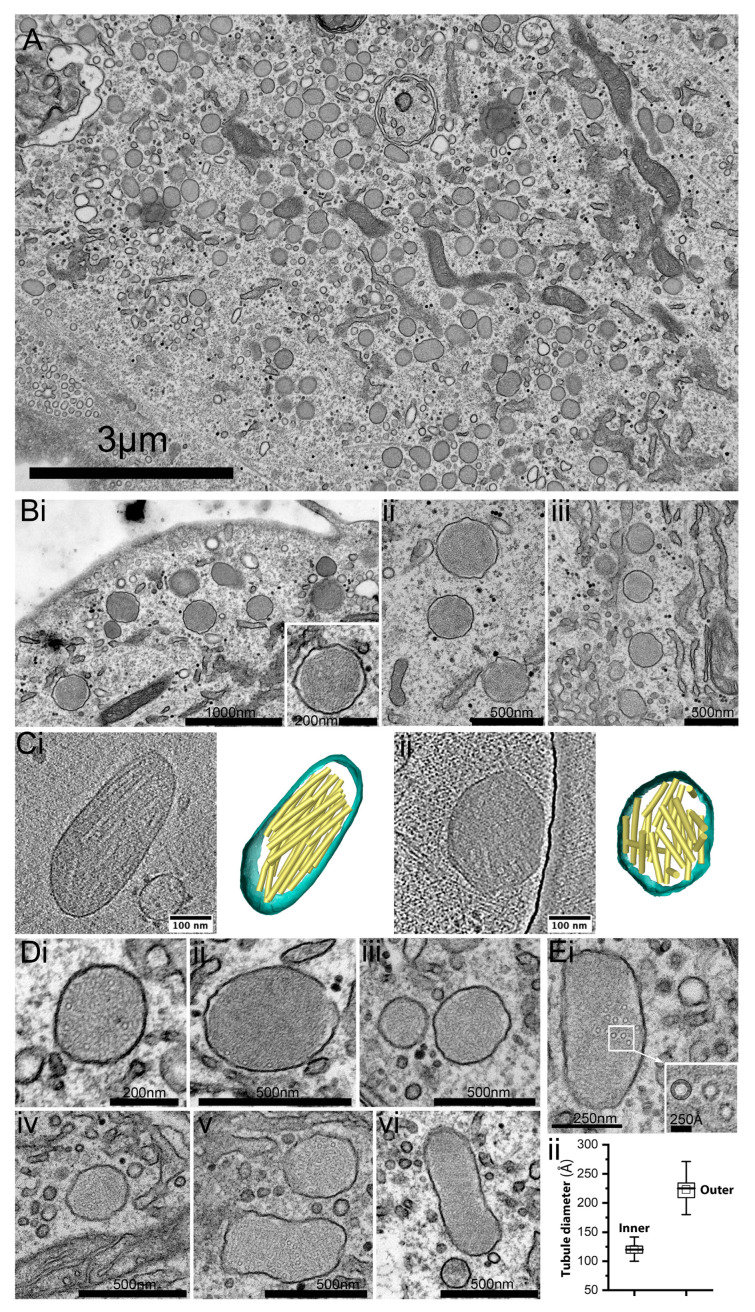
HCMEC_D_ contain numerous rounded WPBs. (**A**) A representative en-face thin section (~60 nm) transmission electron micrograph of a single HCMEC_D_ showing numerous rounded WPBs as electron-dense, membrane-bound organelles within the cell cytoplasm. (**Bi**–**iii**) Examples of rounded WPBs at higher magnification. (**Ci**,**ii**) 3D reconstructions of tubule structures in shortened (left) and rounded (right WPBs) in HCMEC via cryo-ET. (**Di**–**vi**) Examples of WPBs showing tubule-like structures embedded within the luminal material. (**Ei**,**ii**) Quantification of the inner and outer dimensions of tubule-like structures (n = 80 tubules). Whisker box; 25–75% range, inner box; mean, error bars; range 1.5IQR, horizontal bar; median. HCMEC; human cardiac microvascular endothelial cells (_D_; dilated cardiomyopathy). WPBs; Weibel–Palade bodies. Cryo-EM; cryo-electron microscopy. Cryo-ET; cryo-electron tomography.

**Figure 4 ijms-24-04553-f004:**
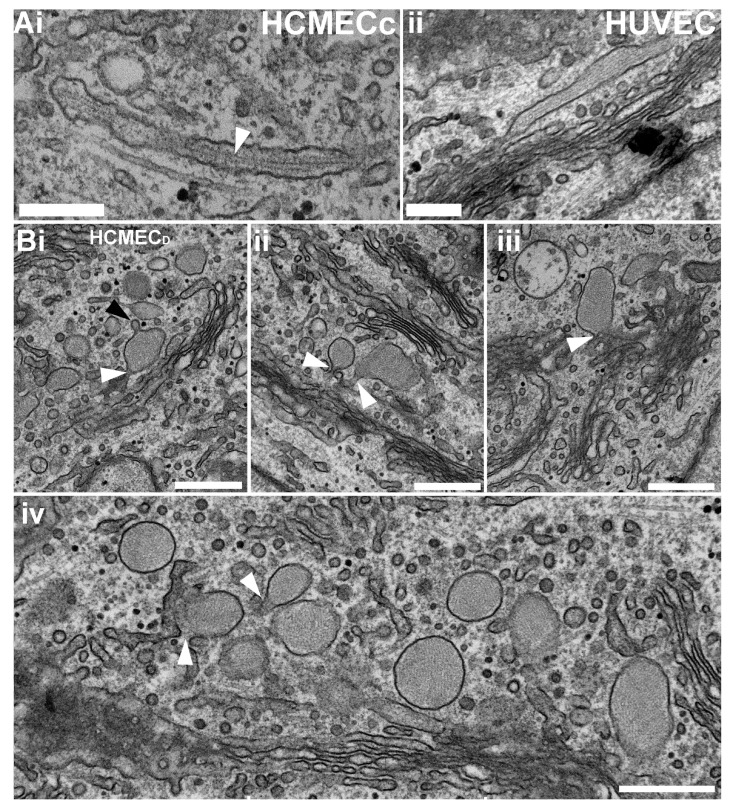
Rounded WPBs in HCMEC_D_ form at the trans-Golgi network. (**Ai**,**ii**) En-face thin section transmission electron micrographs (TEMs) of HCMEC_C_ (**Ai**) and HUVEC (**Aii**) showing mature WPBs with the typical rod-like shape and internal tubules (arrow in (**Ai**)) and striations (**Aii**). (**Bi**–**iv**) Examples of en-face thin section TEMs of HCMEC_D_ showing rounded WPBs connected (arrows) to a reticular network of Golgi membrane stacks. Scale bars: 250 nm. HCMEC; human cardiac microvascular endothelial cells (_C_; control, _D_; dilated cardiomyopathy). WPBs; Weibel–Palade bodies. TEM; transmission electron microscopy.

**Figure 5 ijms-24-04553-f005:**
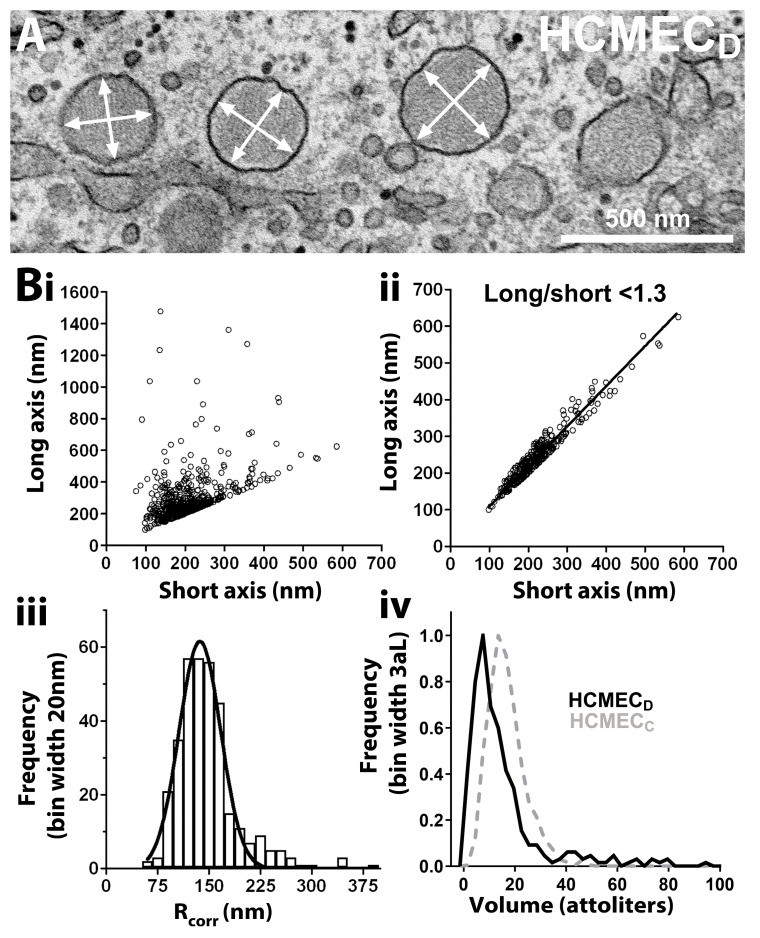
Estimated volumes for WPBs in HCMEC_D_ and HCMEC_C_. (**A**) En-face thin section transmission electron micrograph of HCMEC_D_ with rounded WPBs; white arrows indicate long or short axis of organelles. (**Bi**) Scatter plot of the long vs. short axis lengths of HCMEC_D_ WPBs measured from electron micrographs (n = 54 EM images, 4 independent experiments) and (**Bii**), HCMEC_D_ WPBs with a long:short axis ratio < 1.3 (i.e., approximately spherical in shape, 337/571 WPBs). (**Biii**) Histogram of the radius of HCMEC_D_ WPBs in (**Bii**) corrected for non-diametric sectioning (R_corr)_. (**Biv**) Frequency histogram of the calculated volumes for WPBs in (**Bii**,**iii**). Dashed line in iv is the calculated volume distribution for rod-shaped WPBs in HCMEC_C_, determined as described in Section 4.3 (n = 1231 WPBs, 28 cells, 2 independent experiments). HCMEC; human cardiac microvascular endothelial cells (_C_; control, _D_; dilated cardiomyopathy). WPBs; Weibel–Palade bodies.

**Figure 6 ijms-24-04553-f006:**
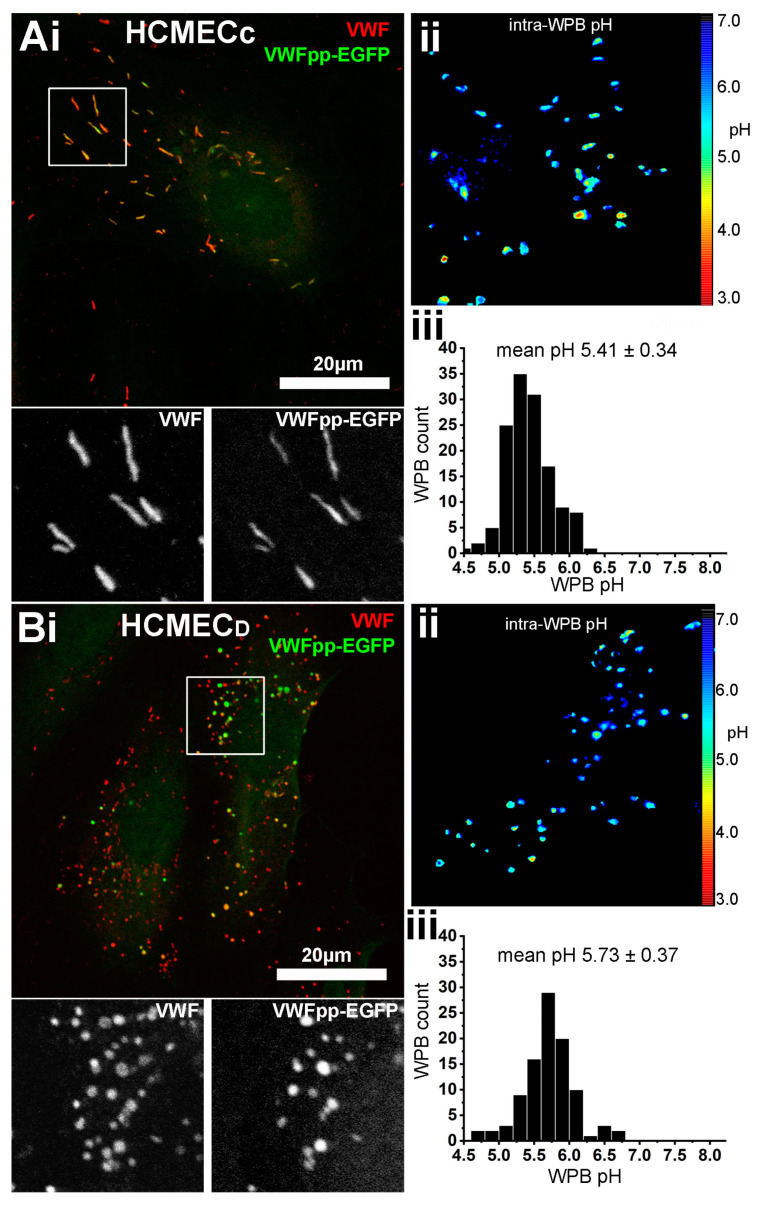
The Intra-luminal pH of WPBs in HCMEC_D_ is marginally less acidic than those of HCMEC_C_. HCMEC_C_ (**Ai**) and HCMEC_D_ (**Bi**) expressing VWFpp-EGFP to specifically label WPBs. Regions in (**Ai**,**Bi**) indicated by white boxes are shown in greyscale below the colour images. (**Aii**,**Bii**) Pseudocolour representations of the calculated intra-WPB pH in single cells and (**Aiii**,**Biii**) the distributions of intra-organelle pH for individual WPBs in HCMEC_C_ (n = 135 WPBs, 7 cells, mean pH 5.41 ± 0.02, s.e.m.) and HCMEC_D_ (n = 92 WPBs, 8 cells, mean pH 5.73 ± 0.03, s.e.m.), respectively. WPB pH in HCMEC_C_ versus HCMEC_D_, *p* < 0.0001, unpaired *t*-test, GraphPad Prism. HCMEC; human cardiac microvascular endothelial cells (_C_; control, _D_; dilated cardiomyopathy). VWFpp-eGFP; von Willebrand factor propeptide-eGFP. WPBs; Weibel–Palade bodies.

**Figure 7 ijms-24-04553-f007:**
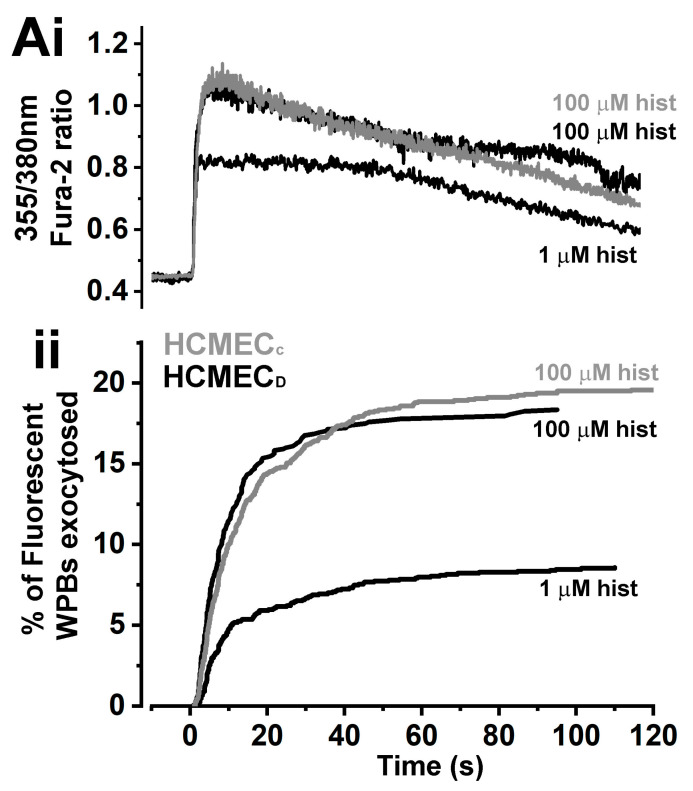
Kinetics of histamine-evoked WPB exocytosis in HCMEC_D_ and HCMEC_C_. (**Ai**) Presentative records of the changes in Fura-2 fluorescence-ratio (355 nm/380 nm) in single HCMEC_D_ and HCMEC_C_ during stimulation with 1.0 or 100 µM histamine as indicated. For comparison, the Fura-2 traces were offset so that the increase in [Ca^2+^]_i_ in each case occurred at time = 0. (**Aii**) Cumulative plots of WPB fusion events normalised by their total number in VWFpp-eGFP expressing HCMEC_D_ (black traces; 1 µM; n = 141 fusion events, 11 cells, 100 µM; n = 196 fusion events, 10 cells) or HCMEC_C_ (grey trace, 100 µM; n = 300 fusion events, 14 cells). HCMEC; human cardiac microvascular endothelial cells (_C_; control, _D_; dilated cardiomyopathy). WPB; Weibel–Palade body. VWFpp-eGFP; von Willebrand factor propeptide-eGFP.

**Figure 8 ijms-24-04553-f008:**
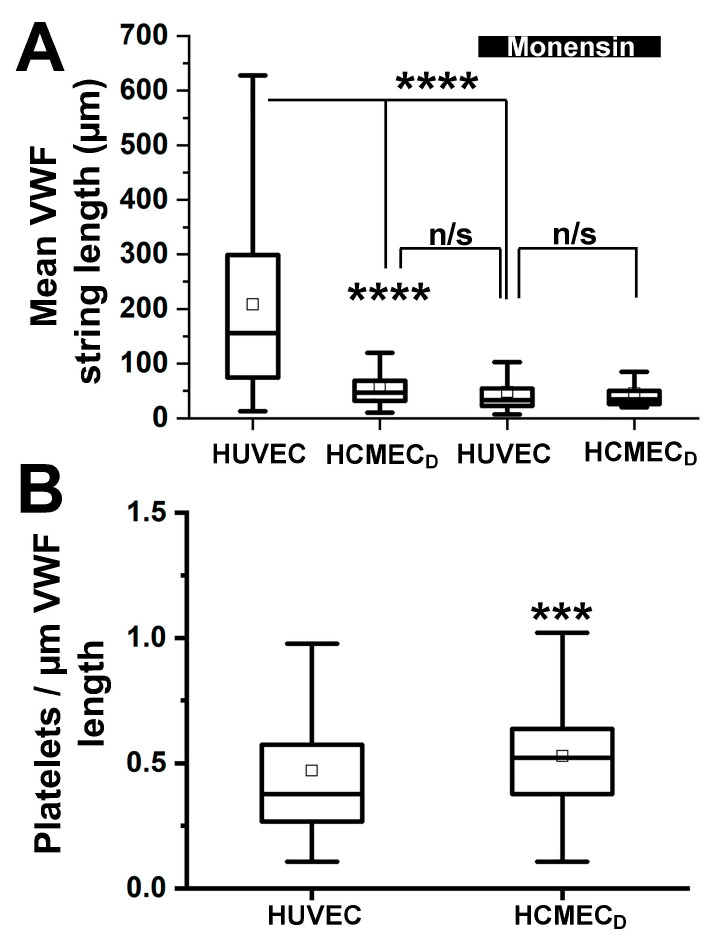
VWFs secreted from HCMEC_D_ form shorter extracellular strings but capture platelets efficiently. (**A**) Whisker plots (25–75% range, inner box; mean, error bars; range 1.5IQR, horizontal bar; median) of the lengths of extracellular strings of VWF produced following histamine-evoked WPB exocytosis in control HUVEC (n = 868 strings from 24 fields of view, 3 independent experiments) or HCMEC_D_ (n = 750 strings from 35 fields of view, 5 separate donors). String lengths after treatment with 10 µM monensin for 2 h: HUVEC; n = 741 strings from 25 fields of view, 2 independent experiments, HCMEC_D_; n = 677 strings from 24 fields of view, 3 separate donors. Olympus IX71 equipped with 20× 0.4NA Ph1 objective, see Section 4.5. (**B**) Number of platelets bound per 1 µm length of VWF secreted from HUVEC (n = 139 strings, 12 fields of view, 3 separate experiments) or HCMEC_D_ (n = 154 strings, 16 fields of view, 3 separate donors). Olympus IX71 equipped with 40× 0.4NA Ph2 objective). (**A**) ****; *p* < 0.0001, one-way ANOVA, GraphPad Prism. (**B**) ***; *p* = 0.003, *t*-test, GraphPad Prism. n/s; no significant difference. WPB; Weibel–Palade body. VWF; von Willebrand factor. HCMEC; human cardiac microvascular endothelial cells (_C_; control, _D_; dilated cardiomyopathy). HUVEC; human umbilical vein endothelial cells. WPBs; Weibel–Palade bodies.

## Data Availability

Source data is provided in MS Excel format accompanying this manuscript. The figures for which source data is available are: Main text Figure 3(Eii), Figure 5, Figure 6, Figure 7, Figure 8 and Appendix A. The code for the ImageJ macro used to calculate intra-organelle pH in organelles containing pH sensors (e.g., eGFP) is provided; contact J.E.M, Justin.Molloy@crick.ac.uk for further information.

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
