# Peer review of "Altered Storage and Function of von Willebrand Factor in Human Cardiac Microvascular Endothelial Cells Isolated from Recipient Transplant Hearts"

_ijms, 2023, doi:10.3390/ijms24054553_

Round 1
Reviewer 1 Report
The study by Meli et al. is interesting and well-conducted. I have several comments that will in my opinion improve this paper:
1. The introduction section should be revised. The authors should briefly introduce us to the topic, address the problem and state the aim (rather then the results).
2. Minor point, figures should have abbreviations listed (as well as statistical test used, when applicable), as these represent independent parts of the manuscript.
3. Perhaps the biggest weakness of the present study is the discussion. The authors have gathered a lot of important informations (covering in my opinion almost all relevant aspects of WPB function), yet they failed to present proper conclusions, or better say implications of their findings. In addition, a reflection on clinical application would be appropriate.
4. Study bears several limitations that were not noted at the end of discussion.
5. Minor point, it says that data is presented as mean and SEM, yet in figures I see box and whiskers plot that hopefully represent median, IQR and min/max - given that this is more appropriate way of presenting these data than mean and SEM.
6. Minor point, the authors should perhaps revise the number of autocitations.
Author Response
We thank Reviewer 1 for the positive comments on our manuscript and are grateful for highlighting the minor issues. We have made the corrections and they are visible in the revised manuscript in Review mode. We feel that the helpful comments and suggestions have greatly improved the MS.
- We have revised the introduction to make it more clear what the main question was, our aims for the study, and how we tested our hypothesis. We have removed references to the results.
- We have listed abbreviations in each figure legend and included details of the statistical tests used.
- We have modified the discussion to provide some conservative conclusions and possible implications of our study as well as to highlight possible clinical applications, such as current efforts to explore pharmacological tools to perturb VWF trafficking and storage as a means to limit or prevent inappropriate coagulation in different disease conditions.
- As requested we have included a brief section at the end of the discussion to indicate some of the study limitations.
- Graphical data is presented as Whisker box plots, we have specified the software used and plot parameters in the methods section “Statistical analysis” and plot parameters in the appropriate figure legends (Figure 3 and the new Figure 8).
- Our use of autocitation covers only essential technical information, specific reagents or background information that provides the first report of specific evidence directly relevant to current work.
Reviewer 2 Report
In general, the authors have performed an interesting study about the ultrastructural changes associated with vWF and its cellular storage. Indeed, research about the ultrastructure of several organelles is fascinating and might elucidate the molecular pathways of function. Nevertheless, I have the following comments to suggest:
1. In the introduction please highlight in the last paragraph the scope of this article. Please, do not present the results of this study.
2. Please, present the methods after the introduction and highlight the ethical considerations of this study in a separate paragraph in the methods. Also, describe in details the statistical analysis that you performed.
3. You could also present your results in a more comprehensive way and provide a detailed support of them in the discussion.
4. Also, please have a second look of the manuscript for minor syntax errors.
Author Response
We thank Reviewer 2 for the positive comments on our manuscript. Thank you also for suggesting we modify the way we present the results to make it more comprehensive and flow better, and to link the results to the discussion better. Your helpful comments and suggestions have greatly improved the MS.
- We have revised the introduction to make it more clear what the scope of the paper was, our aims and how we tested our hypothesis. We have removed references to the results.
- On advice from the editorial office we have for kept the Materials and Methods Section in its current place after the discussion, and added a brief section for “Ethical considerations” and a section for “Statistical analysis” in which we briefly describe the analysis software and statistical tests used in the study.
- Thank you for suggesting we modify the way we present the results. We have re-structured the results to provide separate sections (with sub headings) dealing with where rounded WPBs form (Rounded WPBs of HCMECD form at the trans-Golgi network with disordered VWF tubules), estimation of WPB size (Rounded WPBs of HCMECD are smaller, by volume, than rod-shaped WPBs of HCMECC), and have subdivided the section dealing with WPB exocytosis and VWF string formation in to two separate sections. As part of this we have moved EM images showing VWF tubules in elongated newly forming WPBs in HCMECc and HUVEC (formerly Figure S2Aii and Bii) into a revised Figure 4 (panels 4Ai-ii) so a more ready comparison can be made of the defect in new WPB formation in HCMECD, we have moved panel C from Figure S2 into Figure 5 (new panel A) to help visualised how size measurements were made for WPB volume calculations, and we have split Figure 7 into two sections, one dealing with WPB exocytosis (new Figure 7), and the second with VWF string lengths and platelet capture (new Figure 8) to help with the flow of the results. We have changed the discussion so we link and refer to each specific data figure or result as we move through the narrative.
- We have checked for syntax errors, hopefully we have picked up most of them.